# Ulnar Nerve Dislocation and Subluxation from the Cubital Tunnel Are Common in College Athletes

**DOI:** 10.3390/jcm10143131

**Published:** 2021-07-15

**Authors:** Keisuke Tsukada, Youichi Yasui, Jun Sasahara, Yasuaki Okawa, Takumi Nakagawa, Hirotaka Kawano, Wataru Miyamoto

**Affiliations:** 1Department of Orthopaedic Surgery, Teikyo University School of Medicine, 2-11-1 Kaga, Itabashi-ku, Tokyo 173-8605, Japan; k.r.k.tsuka@gmail.com (K.T.); j.sasa@me.com (J.S.); takumin-tky@umin.ac.jp (T.N.); hkawano-tky@umin.net (H.K.); miyamoto@med.teikyo-u.ac.jp (W.M.); 2Institute of Sports Science & Medicine, Teikyo University, 359 Otsuka, Hachiouji-shi, Tokyo 192-0395, Japan; bnep2009@gmail.com

**Keywords:** ulnar nerve dislocation, ulnar nerve subluxation, ulnar nerve hypermobility, cubital tunnel, ulnar neuropathy, athletes, ultrasonography

## Abstract

Background: Hypermobility of the ulnar nerve from the cubital tunnel reportedly occurs in healthy people without symptoms of ulnar neuropathy. However, the occurrence rate in athletes is unknown. We examined the occurrence rate of ulnar nerve hypermobility using ultrasonography, symptoms, and physical findings in athletes and compared the results of four types of sports. Methods: Medical charts of college athletes competing in baseball, rugby, soccer, and long-distance running between March and November 2018 were retrospectively examined. Dynamic evaluation of the ulnar nerve was performed using ultrasonography and categorized as Types N, S, and D respectively, indicating normal position, subluxation, and dislocation. Subjective and objective findings were evaluated. Results: The present study included 246 male athletes (mean age, 19.7 years; 492 elbows) including 46% Type D, 29.8% Type S, and 24.2% Type N. Subjective findings showed pain and dysesthesia in 9% and 4.5% of participants, respectively, whereas objective findings showed Tinel sign in 6%, nerve tension test in 1.3%, Froment’s sign in 0.5%, and weakness of strength of opponens digiti minimi muscle in 8% of patients with Types D and S. Conclusions: There was a high-frequency hypermobility of the ulnar nerve from the cubital tunnel with or without subjective and objective findings in college athletes.

## 1. Introduction

Hypermobility of the ulnar nerve from the cubital tunnel in the flexion position of the elbow joint may be observed in healthy people without symptoms of ulnar neuropathy, and its frequency among healthy people varies [1,2,3,4,5]. Competitive athletes may have a higher rate of hypermobility than healthy people due to the frequent use of their upper extremities during athletic activities, and this rate may be significantly higher in athletes who engage in activities that involve heavy use of their upper extremities.

Reportedly, a significantly greater degree of movement of the ulnar nerve in patients with ulnar neuropathy at the elbow than in healthy people was determined by ultrasonography [6]. However, the relationship between hypermobility of the ulnar nerve at the elbow joint and ulnar neuropathy has not been fully clarified. Furthermore, while several studies have included healthy people or patients with ulnar neuropathy to investigate the occurrence rate of hypermobility of the ulnar nerve, no studies have evaluated the occurrence rate in athletes [1,2,3,4,5,6].

The hypothesis of the present study was that competitive athletes were likely to have hypermobility of the ulnar nerve compared with athletes who did not routinely use their upper extremities. The present study aimed to verify this hypothesis by examining the occurrence rate of ulnar nerve hypermobility using ultrasonography, symptoms, and physical findings in competitive athletes at athletic clubs within a single university and comparing the results among athletic events.

## 2. Materials and Methods

### 2.1. Participants

Charts from medical check-ups of competitive athletes who belonged to university athletic clubs between March and November 2018 were retrospectively reviewed. The athletic clubs included baseball, rugby, soccer, and long-distance running. Inclusion criteria were male athletes with no history of fracture or surgery in their upper extremities.

### 2.2. Evaluation of Ulnar Nerve and Subjective Findings

Dynamic evaluation of ulnar nerve displacement was performed with participants in the sitting position using an ultrasonography (Noblus; Hitachi Ltd., Tokyo, Japan). All examinations were performed by one board-certified orthopaedic surgeon who was an expert in ultrasonographic examination of musculoskeletal disorders. During dynamic evaluation, participants were asked to extend their elbows with forearm supination, then flex gradually until they reached the full flection position. The position of the ulnar nerve at full flexion of the elbow joint, which was observed using ultrasonography, was classed into the following three groups as previously described by Okamoto et al. [1]: Type N, normal position (ulnar nerve was into the cubital tunnel and not on the medial epicondyle); Type S, subluxation (ulnar nerve was on the tip of the medial epicondyle); and Type D, dislocation (ulnar nerve exceeded and positioned anteriorly to tip of the medial condyle) (Figure 1).

Subjective findings included asking the participants whether they experienced medial elbow pain and dysesthesia in the ulnar half of the ring and little fingers. Tinel sign of the cubital tunnel, nerve tension test [7], Froment’s sign, and strength of the opponens digiti minimi muscle were evaluated as objective findings. The strength of the opponens digiti minimi muscle was evaluated using a pulling test in which participants were instructed to pinch a thin paper between the tip of the thumb and the little finger, and the examiner pulled the paper. The strength of opponens digiti minimi muscle was considered weak if the paper was easily pulled out.

### 2.3. Evaluation

The rates of Types D, S, and N were examined in each club, and the rates of elbows with hypermobility of the ulnar nerve (Types D and S) were compared among clubs. We also compared the frequency of bilateral, only dominant, and only non-dominant cases in participants with hypermobility of the ulnar nerve. Clinical evaluation involved comparing the frequency of subjective and objective findings in Types D and S among clubs. Furthermore, the rate of elbows accompanied by the push-out of the ulnar nerve by the triceps medial head from the cubital tunnel, which was observed using sonographic assessment, was compared among clubs (Figure 2). Finally, we evaluated the rate of Type D, S, and N, subjective and objective findings, and the rate of push-out of the ulnar nerve by the triceps medial head from the cubital tunnel in clubs that involve heavy upper-extremity use (baseball, rugby) as Group H, and in clubs that do not involve heavy upper-extremity use (soccer, long-distance running) as Group L.

### 2.4. Statistical Analysis

Descriptive statistics were presented as mean, standard deviation (SD), numbers (n), and percentage (%). The main analysis involved comparing the occurrence rate of hypermobility and push-out of the ulnar nerve by the triceps between groups H and L in the right and left elbows. JMP15 (SAS Institute Inc., Cary, NC, USA) was used for analysis and Pearson’s chi-square test was performed, with statistical significance set at *p* ≤ 0.001. In addition, sensitivity analysis was performed by comparing the occurrence rate of hypermobility and push-out of the ulnar nerve by the triceps within each group. Pearson’s chi-square test was performed with statistical significance set at *p* ≤ 0.001.

## 3. Results

A total of 246 male athletes (mean age 19.7 years, 492 elbows) were included in the present study. The number of participants in each club was as follows: baseball, 60 (120 elbows); rugby, 63 (126 elbows); soccer, 62 (124 elbows); and long-distance running, 61 (122 elbows). There were 226 Type D elbows (46%), 147 Type S elbows (29.8%), and 119 Type N elbows (24.2%) (Table 1).

Hypermobility of the ulnar nerve at the cubital tunnel (Types D and S) was more frequent in baseball (85%, 51 right elbows and 82%, 49 left elbows) and rugby athletes (91%, 57 right elbows and 92%, 58 left elbows) than in soccer athletes (73%, 45 right elbows and 63%, 39 left elbows) and long-distance running athletes (56%, 34 right elbows and 66%, 40 left elbows). The frequency was significantly higher in group H (88%, 108 right elbows and 87%, 107 left elbows) than group L (64%, 79 right elbows and 64%, 79 left elbows) (*p <* 0.001), but there was no significant difference in frequency within each group (Table 2).

Most of the participants with ulnar nerve hypermobility were bilateral: 88% in baseball, 98% in rugby, 83% in soccer, and 80% in long-distance running (Table 3).

Evaluation of the subjective findings revealed pain in 9% and 5% (34 elbows) and dysesthesia and in 4.5% (17 elbows). Evaluation of the objective findings showed Tinel sign of the cubital tunnel in 6% (22 elbows), nerve tension test in 1.3% (5 elbows), Froment’s sign in 0.5% (2 elbows), and weakness of strength of opponens digiti minimi muscle in 8% (29 elbows) in elbows with Types D and S (Table 4 and Table 5).

Push-out of the ulnar nerve by the triceps long head from the cubital tunnel was observed using sonographic examination and was more frequent in athletes who played baseball (75%, 38 right elbows and 69%, 34 left elbows) and rugby (79%, 45 right elbows and 82%, 48 left elbows) than in those who took part in soccer (58%, 26 right elbows and 47%, 20 left elbows) and long-distance running (29%, 10 right elbows and 35%, 14 left elbows). The frequency was significantly higher in group H (79%, 83 right elbows and 77%, 82 left elbows) than in group L (45%, 36 right elbows and 41%, 34 left elbows) (*p <* 0.001), but there was no significant difference in the frequency within each group (Table 6).

## 4. Discussion

The main finding of the present study was that 75.8% of university competitive athletes had hypermobility of the ulnar nerve at the cubital tunnel (Types D and S) with or without subjective and objective findings. Previous studies have examined the rate of ulnar nerve hypermobility at the cubital tunnel in healthy individuals using various procedures [1,2,3,4,5,8,9]. Okamoto et al. examined 200 elbows in 100 healthy volunteers using ultrasonography and showed that 47% of healthy volunteers had ulnar nerve hypermobility at the cubital tunnel [1]. Ozturk et al. studied the ultrasonographic appearance of the ulnar nerve in the cubital tunnel in 212 elbows of healthy volunteers, which is the largest number of participants previously studied, and showed an ulnar nerve hypermobility rate of 31.6% [3]. In 2019, Cornelson et al. used ultrasonography of ulnar nerve instability in 84 elbows of healthy individuals and reported that ulnar nerve instability was observed in 56%, which is the highest rate of hypermobility of the ulnar nerve at the cubital tunnel reported using this method [5]. Recent studies have evaluated hypermobility of the ulnar nerve at the cubital tunnel using magnetic resonance imaging (MRI) [8,9]. Kawahara et al. assessed 100 healthy elbows of 50 volunteers using MRI and showed that 49% of healthy elbows had hypermobility of the ulnar nerve at the level of the cubital nerve [9].

Two previous studies performed ultrasonographic evaluation to compare the incidence of ulnar nerve hypermobility at the cubital tunnel between patients with ulnar neuropathy and healthy controls [6,10]. Van Den Berg et al. performed an ultrasonographic comparative study in 342 patients who suffered from ulnar neuropathy and 70 healthy controls, and reported that there was no significant intergroup difference in occurrence of ulnar nerve hypermobility [10]. Yang et al. performed ultrasonographic evaluation of 26 patients with ulnar neuropathy and 13 control participants, and revealed that there was significantly greater displacement of the ulnar nerve to the medial epicondyle at the inlet of the cubital tunnel in the patients with ulnar neuropathy during elbow extension and flexion [6]. The present study found a rate of hypermobility of the ulnar nerve at the cubital tunnel of 75.8% in competitive athletes, which is high compared with previous reports. One of the reasons for this finding may be the routine use of the upper extremities during the athletic activities studied.

Previous studies have considered the triceps medial head as a factor involved in hypermobility of the ulnar nerve at the cubital tunnel [11,12,13,14,15,16,17,18]. Flexion of the elbow commonly causes entry of the triceps medial head into the proximal aspect of the cubital tunnel, and this phenomenon was correlated with hypermobility of the ulnar nerve. Previous studies used the term “snapping” to describe the triceps medial head over the medial epicondyle anteriorly [11,12,13]. Snapping of the triceps medial head was first described in 1970 [11]. In 1998, Spinner et al. reported a case series of 17 patients with recurrent dislocation of the ulnar nerve accompanied by snapping of the medial head of the triceps [12]. In 2011, Molnar et al. reported a case of dislocation of the ulnar nerve in an elite wrestler and concluded that the prominent medial head of the triceps further pushed out the ulnar nerve from the sulcus in athletes with well-developed upper-limb muscles [17]. Michael and Young focused on hypertrophy of the triceps as a contributing factor for ulnar nerve luxation and compared the frequency between control and bodybuilder groups [18]. They showed a significantly higher frequency of the ulnar nerve luxation in bodybuilders. In the present study, Types D and S were significantly more frequent in baseball and rugby athletes than in soccer and long-distance running athletes. Furthermore, push-out of the ulnar nerve by the triceps medial head from the cubital tunnel, which was observed using ultrasonographic examination, was significantly more frequent in baseball and rugby athletes than in soccer and long-distance running athletes. These results indicated that athletes who engaged in athletic activities that involve heavy use of upper extremities were more likely to have hypermobility of ulnar nerve compared with those who did not routinely use their upper extremities. These findings are supported by the results of previous studies [17,18].

The present study also found that the subjective and objective findings in Types D and S were medial elbow pain (9%) and weakness of strength of opponens digiti minimi muscle (8%). Although these were rare, athletes with ulnar nerve hypermobility at the cubital tunnel may have substantial findings of ulnar nerve neuropathy, and cautious observation is required for such cases. The present study was cross-sectional in design, which is not suitable for determining the causal relationship between subjective and objective findings. Further studies are required to determine the association between ulnar dislocation and neurological symptoms. Intervention may be required for cases with ulnar nerve dislocation.

The present study has several limitations. First, we included only male athletes, and the outcomes from this study may differ in female athletes. Second, we did not perform MRI evaluation for the location of the ulnar nerve at the cubital tunnel, which may suggest more objective outcomes. Furthermore, this was a cross-sectional study and we were unable to determine the causal relationship between the subjective and objective findings. In addition, there was no significant difference in the frequency and clinical characteristics of ulnar nerve hypermobility between right and left arms or between dominant and non-dominant arms. However, clinical differences may occur in other sports with greater right–left or dominant arm loads. Further research is needed in this regard.

## 5. Conclusions

A total of 75.8% of college athletes who engaged in competitive levels had hypermobility of the ulnar nerve at the cubital tunnel with or without subjective and objective findings. Our findings show a high frequency compared with the results of previous studies that examined hypermobility of the ulnar nerve for healthy participants. Hypermobility of the ulnar nerve was significantly more frequent in baseball and rugby athletes than in soccer players and long-distance runners. Furthermore, push-out of the ulnar nerve by the triceps medial head from the cubital tunnel, which was observed using ultrasonographic examination, was significantly more frequent in baseball and rugby athletes than in soccer players and long-distance runners. These findings revealed that athletes who engaged in athletic activities that involve heavy use of the upper extremities were more likely to have hypermobility of the ulnar nerve compared with athletes who did not routinely use their upper extremities.

## Figures and Tables

**Figure 1 jcm-10-03131-f001:**
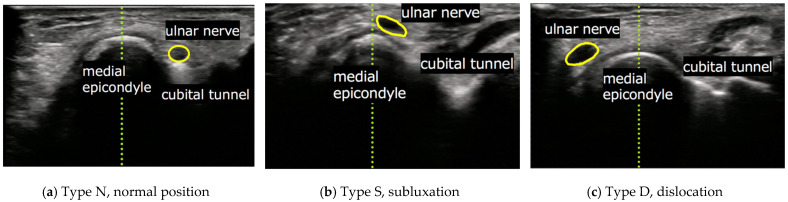
Short axis images at the cubital tunnel by ultrasonography. (**a**) Type N, normal position (ulnar nerve was into the cubital tunnel); (**b**) Type S, subluxation (ulnar nerve was on the tip of the medial epicondyle); and (**c**) Type D, dislocation (ulnar nerve exceed and positioned anteriorly to tip of the medial condyle).

**Figure 2 jcm-10-03131-f002:**
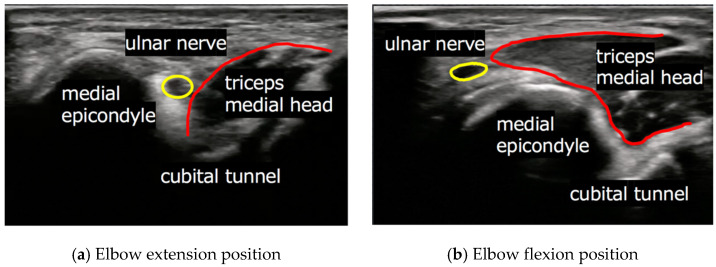
Short-axis images at the cubital tunnel using ultrasonography. (**a**) Elbow extension position and (**b**) elbow flexion position: push-out of the ulnar nerve by the triceps medial head from the cubital tunnel.

**Table 1 jcm-10-03131-t001:** Demographic data.

	Baseball	Rugby	Soccer	Long-Distance Running	Total
*n*	60	63	62	61	246
Age, mean ± SD	19.6 ± 1.1	19.8 ± 1.2	19.7 ± 1.0	19.5 ± 1.2	19.7 ± 1.1
Male (*n*)	60	63	62	61	246
Elbows (*n*)	120	126	124	122	492
Right (*n*)/Left (*n*)	60/60	63/63	62/62	61/61	246/246
Type D (*n*, %)	29 (48)/33 (55)	35 (56)/42 (67)	26 (42)/24 (39)	17 (28)/20 (33)	107 (44)/119 (49)
Type S (*n*, %)	22 (37)/16 (27)	22 (35)/16 (25)	19 (31)/15 (24)	17 (28)/20 (33)	80 (33)/67 (27)
Type N (*n*, %)	9 (15)/11 (18)	6 (9)/5 (8)	17 (27)/23 (37)	27 (44)/21 (34)	59 (23)/60 (24)

SD, standard deviation.

**Table 2 jcm-10-03131-t002:** Frequency of hypermobility of the ulnar nerve among groups H and L.

		Types D and S	Type N	*p*-Value
**Right**	**Group**			<0.001 †
	Group H (*n*, %)	108 (88)	15 (12)	
	Baseball (*n*, %)	51 (85)	9 (15)	0.15 ‡
	Rugby (*n*, %)	57 (91)	6 (9)	
	Group L (*n*, %)	79 (64)	44 (36)	
	Soccer (*n*, %)	45 (73)	17 (27)	0.05 ‡
	Long-distance running (*n*, %)	34 (56)	27 (44)	
**Left**	**Group**			<0.001 †
	Group H (*n*, %)	107 (87)	16 (13)	
	Baseball (*n*, %)	49 (82)	11 (16)	0.03 ‡
	Rugby (*n*, %)	58 (92)	5 (9)	
	Group L (*n*, %)	79 (64)	44 (36)	
	Soccer (*n*, %)	39 (63)	23 (37)	0.75 ‡
	Long-distance running (*n*, %)	40 (66)	21 (34)	

Type N, normal position (ulnar nerve was into the cubital tunnel and not on the medial epicondyle). *n*, number. † *p*-value comparing the occurrence rate of hypermobility between groups H and L. ‡ *p*-value comparing the occurrence rate of hypermobility within each group.

**Table 3 jcm-10-03131-t003:** Frequency of bilateral, only dominant, or only non-dominant cases among participants with hypermobility of the ulnar nerve.

	Baseball	Rugby	Soccer	Long-Distance Running
Types D and S (*n*)	53	58	46	41
Bilateral (*n*, %)	47 (88)	57 (98)	38 (83)	33 (80)
Only dominant (*n*, %)	4 (8)	0 (0)	7 (15)	3 (8)
Only non-dominant (*n*, %)	2 (4)	1 (2)	1 (2)	5 (12)

**Table 4 jcm-10-03131-t004:** Frequency of subjective symptoms among elbows with Types D and S.

	Baseball	Rugby	Soccer	Long-Distance Running	Total
Types D and S (*n*)	100	115	84	74	373
Right/Left (*n*)	51/49	57/58	45/39	34/40	187/186
Pain (*n*, %)	15 (29)/5 (10)	6 (11)/3 (5)	3 (7)/2 (5)	0 (0)/0 (0)	24 (13)/10 (5)
Dysesthesia (*n*, %)	4 (8)/1 (2)	5 (9)/3 (5)	2 (4)/2 (5)	0 (0)/0 (0)	11 (6)/6 (3)

**Table 5 jcm-10-03131-t005:** Frequency of objective findings among elbows with Types D and S.

	Baseball	Rugby	Soccer	Long-Distance Running	Total
Types D and S (*n*)	100	115	84	74	373
Right/Left (*n*)	51/49	57/58	45/39	34/40	187/186
Tinel sign (*n*, %)	3 (6)/1 (2)	6 (10)/4 (6)	2 (4)/2 (5)	2 (6)/2 (6)	13 (7)/9 (5)
NTT (*n*, %)	1 (2)/0 (0)	1 (2)/2 (3)	1 (2)/0 (2)	0 (0)/0 (0)	3 (2)/2 (1)
Froment’s sign (*n*, %)	0 (0)/0 (0)	0 (0)/2 (3)	0 (0)/0 (0)	0 (0)/0 (0)	0 (0)/2 (1)
Weakness of opponens digiti minimi muscle (*n*, %)	2 (4)/3 (6)	8 (14)/11 (19)	3 (6)/2 (5)	0 (0)/0 (0)	13 (7)/16 (9)

NTT, nerve tension test.

**Table 6 jcm-10-03131-t006:** Frequency of push-out of the ulnar nerve by the triceps from the cubital tunnel among groups H and L.

		Push-Out of the Ulnar Nerve by the Triceps	Others	*p*-Value
**Right**	**Group**			<0.001 †
	Group H (*n*, %)	83 (79)	25 (21)	
	Baseball (*n*, %)	38 (75)	13 (25)	0.58 ‡
	Rugby (*n*, %)	45 (79)	12 (21)	
	Group L (*n*, %)	36 (45)	43 (55)	
	Soccer (*n*, %)	26 (58)	19 (43)	0.01 ‡
	Long-distance running (*n*, %)	10 (29)	24 (71)	
**Left**	**Group**			<0.001 †
	Group H (*n*, %)	82 (77)	25 (23)	
	Baseball (*n*, %)	34 (69)	15 (31)	0.10 ‡
	Rugby (*n*, %)	48 (82)	10 (18)	
	Group L (*n*, %)	34 (41)	49 (59)	
	Soccer (*n*, %)	20 (47)	23 (53)	0.28 ‡
	Long-distance running (*n*, %)	14 (35)	26 (65)	

† *p*-value comparing the occurrence rate of push-out of the ulnar nerve by the triceps between groups H and L. ‡ *p*-value comparing the occurrence rate of push-out of the ulnar nerve by the triceps within each group.

## Data Availability

The data presented in this study are available on request from the corresponding author.

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
