# Peer review of "Ulnar Nerve Dislocation and Subluxation from the Cubital Tunnel Are Common in College Athletes"

_jcm, 2021, doi:10.3390/jcm10143131_

Round 1
Reviewer 1 Report
Thank you for allowing me to review this interesting work. It is an excellent manuscript.
It is a retrospective cross-sectional study focusing on ulnar nerve dynamic evaluation by ultrasonotherapy in male college athletes (baseball, rugby, soccer, and long-distance running). In this cohort, 75.8% of university athletes had hypermobility of the ulnar nerve at the cubital tunnel (Dislocated or Subluxation). Hypermobility of the ulnar nerve was significantly more frequent in baseball and rugby athletes. Furthermore, the triceps medial head's push-out of the ulnar nerve from the cubital tunnel, which was observed using ultrasonographic examination, was significantly more frequent in baseball and rugby athletes. These findings revealed that athletes who engaged in athletic activities that involve heavy use of the upper extremities were likely to have hypermobility of the ulnar nerve compared with athletes who did not routinely use their upper extremities.
The manuscript is well written, easy to read, with clear objectives and methodology, nice tables, and comprehensive figures. Results are original on a consequent cohort with high values of ulnar nerve dislocation compared to literature. The discussion is well conducted, with an interesting hypothesis, such as the medial triceps head. The limits concerning the clinical consequences with ulnar neuropathy of this dislocation/subluxation are well stated, with cautious conclusions.
I have only one comment,
- I would suggest analyzing or discussing the dominant elbow (right-handed or left-handed), especially in baseball players, and comparing it to the other side.
Author Response
Ulnar Nerve Dislocation and Subluxation from the Cubital Tunnel are Common in College Athletes
(jcm-1260659).
Thank you for inviting us to submit a revised version of our manuscript. We greatly appreciate the comments and suggestions of the reviewers. Our point-by-point responses to the comments are detailed below. Changes to the manuscript are shown in bold text.
COMMENTS TO AUTHOR:
<Reviewer 1>
Comments:
Thank you for allowing me to review this interesting work. It is an excellent manuscript.
It is a retrospective cross-sectional study focusing on ulnar nerve dynamic evaluation by ultrasonotherapy in male college athletes (baseball, rugby, soccer, and long-distance running). In this cohort, 75.8% of university athletes had hypermobility of the ulnar nerve at the cubital tunnel (Dislocated or Subluxation). Hypermobility of the ulnar nerve was significantly more frequent in baseball and rugby athletes. Furthermore, the triceps medial head's push-out of the ulnar nerve from the cubital tunnel, which was observed using ultrasonographic examination, was significantly more frequent in baseball and rugby athletes. These findings revealed that athletes who engaged in athletic activities that involve heavy use of the upper extremities were likely to have hypermobility of the ulnar nerve compared with athletes who did not routinely use their upper extremities.
The manuscript is well written, easy to read, with clear objectives and methodology, nice tables, and comprehensive figures. Results are original on a consequent cohort with high values of ulnar nerve dislocation compared to literature. The discussion is well conducted, with an interesting hypothesis, such as the medial triceps head. The limits concerning the clinical consequences with ulnar neuropathy of this dislocation/subluxation are well stated, with cautious conclusions.
I have only one comment,
- I would suggest analyzing or discussing the dominant elbow (right-handed or left-handed), especially in baseball players, and comparing it to the other side.
Response:
Thank you for spending your precious time. Also, we are very honored to have a positive comment. According to your recommendation, we have created a new Table 3 to show the frequency of bilateral, only dominant, or only non-dominant cases among participants with hypermobility of the ulnar nerve. As a result, Table 3 in the initial manuscript was changed to Table 4, Table 4 to 5, and Table 5 to 6.
We also added new details about the right and left sides, according to the suggestion from both reviewers. The frequencies are listed separately for right and left elbows in tables 1, 2, 4, 5, and 6.
Line 90-92:
We also compared the frequency of bilateral, only dominant, and only non-dominant cases in participants with hypermobility of the ulnar nerve.
Line 110-112:
The main analysis involved comparing the occurrence rate of hypermobility and push-out of the ulnar nerve by the triceps between groups H and L in the right and left elbows.
Line 128-135:
Hypermobility of the ulnar nerve at the cubital tunnel (Types D and S) were more frequent in baseball (85%, 51 right elbows and 82%, 49 left elbows) and rugby athletes (91%, 57 right elbows and 92%, 58 left elbows) than in soccer (73%, 45 right elbows and 63%, 39 left elbows) and long-distance run athletes (56%, 34 right elbows and 66%, 40 left elbows). The frequency was significantly higher in group H (88%, 108 right elbows and 87%, 107 left elbows) than group L (64%, 79 right elbows and 64%, 79 left elbows) (P < 0.001), but there was no significant difference in frequency within each group (Table 2).
Line 143-144:
Most of the participants with ulnar nerve hypermobility were bilateral: 88% in baseball, 98% in rugby, 83% in soccer, and 80% in long-distance running (Table 3).
Line 150-154:
Evaluation of the subjective findings revealed pain in 9% (34 elbows) and dysesthesia and in 4.5% (17 elbows). Evaluation of the objective findings showed Tinel sign of the cubital tunnel in 6% (22 elbows), nerve tension test in 1.3% (5 elbows), Froment’s sign in 0.5% (2 elbows), and weakness of strength of opponens digiti minimi muscle in 8% (29 elbows) in elbows with Types D and S (Tables 4 and 5).
Line 164-172:
Push-out of the ulnar nerve by the triceps long head from the cubital tunnel was observed using sonographic examination and was more frequent in athletes who played baseball (75%, 38 right elbows and 69%, 34 left elbows) and rugby (79%, 45 right elbows and 82%, 48 left elbows) than those who took part in soccer (58%, 26 right elbows and 47%, 20 left elbows) and long-distance running (29%, 10 right elbows and 35%, 14 left elbows). The frequency was significantly higher in group H (79%, 83 right elbows and 77%, 82 left elbows) than in group L (45%, 36 right elbows and 41%, 34 left elbows) (P < 0.001), but there was no significant difference in the frequency within each group (Table 6).
Thank you again for your review.
Reviewer 2 Report
This study reports the prevalence of ulnar nerve hypermobility at the cubital tunnel in a cohort of college athletes from different sports. This is an important study showing a high prevalence of ulnar nerve dislocation and subluxation at the elbow in young athletes than previously reported and is important for the prevention of more severe ulnar nerve neuropathy in sportsmen.
Although I endorse the study for publication, there is an important issue that requires attention, which is related with considering each elbow as the unit of analysis. While this may be a common practice that follows previous studies, using each elbow’s data as an observation is a flagrant violation of the independence assumption of the Pearson’s chi square, as it is for several other statistical tests, that unduly increases the sample size. Moreover, by analysing each elbow separately important clinical information is missed, like the presence of bilateral or unilateral ulnar nerve hypermobility or what are the most frequent patterns of ulnar nerve hypermobility alterations between the two body sides and how these patterns relate with the sports discipline. For example, it would be interesting to compare ulnar nerve hypermobility between baseball and rugby, since baseball is characterised by powerful throwing actions and asymmetrical movements of the upper limbs, compared to the more symmetrical actions of rugby. Thereby, the use of elbows as unit of analysis should be avoided based on statistical and clinical reasons and recommend to the authors to revise their statistical analysis.
Minor issues
Page 3, lines 109-114. Frequency data is reported with reference to “participants” when it should say “elbows”. The term “participants” is also ambiguously used in following sentences of the Results section.
Table 4. Where it says “Strength of opponens…” probably it would be more meaningful saying “Weakness of opponens…”.
Author Response
Ulnar Nerve Dislocation and Subluxation from the Cubital Tunnel are Common in College Athletes
(jcm-1260659).
Thank you for inviting us to submit a revised version of our manuscript. We greatly appreciate the comments and suggestions of the reviewers. Our point-by-point responses to the comments are detailed below. Changes to the manuscript are shown in bold text.
COMMENTS TO AUTHOR:
<Reviewer 2>
Comments:
This study reports the prevalence of ulnar nerve hypermobility at the cubital tunnel in a cohort of college athletes from different sports. This is an important study showing a high prevalence of ulnar nerve dislocation and subluxation at the elbow in young athletes than previously reported and is important for the prevention of more severe ulnar nerve neuropathy in sportsmen.
Although I endorse the study for publication, there is an important issue that requires attention, which is related with considering each elbow as the unit of analysis. While this may be a common practice that follows previous studies, using each elbow’s data as an observation is a flagrant violation of the independence assumption of the Pearson’s chi square, as it is for several other statistical tests, that unduly increases the sample size. Moreover, by analyzing each elbow separately important clinical information is missed, like the presence of bilateral or unilateral ulnar nerve hypermobility or what are the most frequent patterns of ulnar nerve hypermobility alterations between the two body sides and how these patterns relate with the sports discipline. For example, it would be interesting to compare ulnar nerve hypermobility between baseball and rugby, since baseball is characterized by powerful throwing actions and asymmetrical movements of the upper limbs, compared to the more symmetrical actions of rugby. Thereby, the use of elbows as unit of analysis should be avoided based on statistical and clinical reasons and recommend to the authors to revise their statistical analysis.
Response:
Thank you for spending your precious time. We appreciate your positive comments on this study. Regarding analyzing both sides of the elbows in the study, we included athletes who were able to engage in sports activities, regardless of whether they had ulnar neuropathy or not. Therefore, bilateral data were collected in the research phase rather than unilateral data. Since the data used in the statistical analysis were nominal scale variables and we would like to avoid multiplicity concerns by multiple analyses, the data were classified into two groups, Group H and Group L, and performed Pearson's Chi-square test. In addition, we added details for the left and right sides and performed the analysis separately for the left and right sides to avoid inadequately increasing the sample size by using bilateral data in the statistical analysis. We have created a new Table 3 to show the frequency of bilateral, only dominant, or only non-dominant cases among participants with hypermobility of the ulnar nerve. As a result, Table 3 in the initial manuscript was changed to Table 4, Table 4 to 5, and Table 5 to 6. In Tables 1, 2, 4, 5, and 6. The frequencies were listed separately for right and left elbows.
Regarding the important clinical information, we agree with your point there may be right/left or dominant arm differences in sports where the muscles of the dominant arm are likely to develop. Although there were no differences between the left and right sides in the present study, we added the comment to the discussion the parts that have been pointed out so far.
Line 90-92:
We also compared the frequency of bilateral, only dominant, and only non-dominant cases in participants with hypermobility of the ulnar nerve.
Line 110-112:
The main analysis involved comparing the occurrence rate of hypermobility and push-out of the ulnar nerve by the triceps between groups H and L in the right and left elbows.
Line 128-135:
Hypermobility of the ulnar nerve at the cubital tunnel (Types D and S) were more frequent in baseball (85%, 51 right elbows and 82%, 49 left elbows) and rugby athletes (91%, 57 right elbows and 92%, 58 left elbows) than in soccer (73%, 45 right elbows and 63%, 39 left elbows) and long-distance run athletes (56%, 34 right elbows and 66%, 40 left elbows). The frequency was significantly higher in group H (88%, 108 right elbows and 87%, 107 left elbows) than group L (64%, 79 right elbows and 64%, 79 left elbows) (P < 0.001), but there was no significant difference in frequency within each group (Table 2).
Line 143-144:
Most of the participants with ulnar nerve hypermobility were bilateral: 88% in baseball, 98% in rugby, 83% in soccer, and 80% in long-distance running (Table 3).
Line 150-154:
Evaluation of the subjective findings revealed pain in 9% (34 elbows) and dysesthesia and in 4.5% (17 elbows). Evaluation of the objective findings showed Tinel sign of the cubital tunnel in 6% (22 elbows), nerve tension test in 1.3% (5 elbows), Froment’s sign in 0.5% (2 elbows), and weakness of strength of opponens digiti minimi muscle in 8% (29 elbows) in elbows with Types D and S (Tables 4 and 5).
Line 164-172:
Push-out of the ulnar nerve by the triceps long head from the cubital tunnel was observed using sonographic examination and was more frequent in athletes who played baseball (75%, 38 right elbows and 69%, 34 left elbows) and rugby (79%, 45 right elbows and 82%, 48 left elbows) than those who took part in soccer (58%, 26 right elbows and 47%, 20 left elbows) and long-distance running (29%, 10 right elbows and 35%, 14 left elbows). The frequency was significantly higher in group H (79%, 83 right elbows and 77%, 82 left elbows) than in group L (45%, 36 right elbows and 41%, 34 left elbows) (P < 0.001), but there was no significant difference in the frequency within each group (Table 6).
Line 248-252:
In this study, there was no significant difference in the frequency and clinical characteristics of ulnar nerve hypermobility between right and left arms or between dominant and non-dominant arms. However, clinical differences may occur in other sports with greater right-left or dominant arm loads. Further research is needed in this regard.
Minor issues
Page 3, lines 109-114. Frequency data is reported with reference to “participants” when it should say “elbows”. The term “participants” is also ambiguously used in following sentences of the Results section.
Response:
Reviewed and revised when it should be participants and when it should be elbows.
Line 128-135:
Hypermobility of the ulnar nerve at the cubital tunnel (Types D and S) were more frequent in baseball (85%, 51 right elbows and 82%, 49 left elbows) and rugby athletes (91%, 57 right elbows and 92%, 58 left elbows) than in soccer (73%, 45 right elbows and 63%, 39 left elbows) and long-distance run athletes (56%, 34 right elbows and 66%, 40 left elbows). The frequency was significantly higher in group H (88%, 108 right elbows and 87%, 107 left elbows) than group L (64%, 79 right elbows and 64%, 79 left elbows) (P < 0.001), but there was no significant difference in frequency within each group (Table 2).
Line 150-154:
Evaluation of the subjective findings revealed pain in 9% (34 elbows) and dysesthesia and in 4.5% (17 elbows). Evaluation of the objective findings showed Tinel sign of the cubital tunnel in 6% (22 elbows), nerve tension test in 1.3% (5 elbows), Froment’s sign in 0.5% (2 elbows), and weakness of strength of opponens digiti minimi muscle in 8% (29 elbows) in elbows with Types D and S (Tables 4 and 5).
Line 164-172:
Push-out of the ulnar nerve by the triceps long head from the cubital tunnel was observed using sonographic examination and was more frequent in athletes who played baseball (75%, 38 right elbows and 69%, 34 left elbows) and rugby (79%, 45 right elbows and 82%, 48 left elbows) than those who took part in soccer (58%, 26 right elbows and 47%, 20 left elbows) and long-distance running (29%, 10 right elbows and 35%, 14 left elbows). The frequency was significantly higher in group H (79%, 83 right elbows and 77%, 82 left elbows) than in group L (45%, 36 right elbows and 41%, 34 left elbows) (P < 0.001), but there was no significant difference in the frequency within each group (Table 6)
Minor issues
Table 4. Where it says “Strength of opponens…” probably it would be more meaningful saying “Weakness of opponens…”.
Response:
We edited the text accordingly. Please check the table 5 which was revised from table 4 in previous version.
Thank you again for your review.